# Effect of Infant and Maternal Secretor Status on Rotavirus Vaccine Take—An Overview

**DOI:** 10.3390/v13061144

**Published:** 2021-06-14

**Authors:** Sumit Sharma, Johan Nordgren

**Affiliations:** Division of Molecular Medicine and Virology, Department of Clinical and Biomedical Sciences, Linköping University, 58183 Linköping, Sweden; sumit.sharma@liu.se

**Keywords:** rotavirus, secretor status, vaccine take, mother, infant

## Abstract

Histo-blood group antigens, which are present on gut epithelial surfaces, function as receptors or attachment factors and mediate susceptibility to rotavirus infection. The major determinant for susceptibility is a functional FUT2 enzyme which mediates the presence of α-1,2 fucosylated blood group antigens in mucosa and secretions, yielding the secretor-positive phenotype. Secretors are more susceptible to infection with predominant rotavirus genotypes, as well as to the commonly used live rotavirus vaccines. Difference in susceptibility to the vaccines is one proposed factor for the varying degree of efficacy observed between countries. Besides infection susceptibility, secretor status has been found to modulate rotavirus specific antibody levels in adults, as well as composition of breastmilk in mothers and microbiota of the infant, which are other proposed factors affecting rotavirus vaccine take. Here, the known and possible effects of secretor status in both infant and mother on rotavirus vaccine take are reviewed and discussed.

## 1. Introduction—Secretor Status and Susceptibility to Rotavirus

Group A rotavirus, hereafter called rotavirus in this review, is the most common cause of severely dehydrating gastroenteritis in children worldwide [1]. The virus is highly infectious, and most children have had several rotavirus infections by the age of five years. Nevertheless, studies have shown that some children are more resistant to infection and disease for particular rotavirus genotypes [2]. This resistance or susceptibility is largely mediated through expression of human histo-blood group antigens (HBGAs), mainly the types controlled by the *FUT2* (secretor), *FUT3* (Lewis), and *ABO* genes, in a rotavirus genotype dependent manner. The protease-activated spike protein VP4, which, together with glycoprotein VP7 (G genotype), elicits neutralization antibodies, is used to define protease sensitive (P) serotypes or genotypes. Cellular attachment and entry, as well as HBGA binding in vitro, is mediated by the VP4 protein (P-genotype), which is post-translationally cleaved into a glycan binding domain and polypeptides [3]. As such, the P-genotypes determine the pattern of genetic susceptibility. Globally, three P-genotypes, namely P[4], P[6], and P[8], are common in humans [2,4], whereas other P-genotypes only occasionally infect humans. These three predominant rotavirus P-genotypes are thus the most clinically relevant is terms of susceptibility to natural rotavirus infections. 

Both in vitro and observational studies have provided strong evidence that secretor (FUT2) and Lewis (FUT3) antigens mediate susceptibility to the predominant genotypes in a P-genotype-dependent manner [2,5]. Positive secretor status has been associated with increased susceptibility to P[8] and P[4] infections, whereas P[6] infections are strongly associated with the Lewis-negative phenotype, independent of secretor status [2]. Some differences exist between studies, mainly in regard to the most common, and most studied, P[8] genotype. For example, a few studies have reported that Lewis-positivity, independent of secretor status, was the major susceptibility factor, while others studies observed that the Lewis b-phenotype present in individuals that are both secretor and Lewis positive was a stronger susceptibility marker than only secretor-positive status [2]. Although less studies for P[4] rotaviruses are available, similar differences between studies have been reported [2]. 

Moreover, an in vitro study [6] showed that structural differences of different P[8] lineages, as well as the Rotarix vaccine, yielded different binding patterns to glycans. For example, Rotarix was different from other P[8] strains of the same lineage I, showing a lack of interaction with the α1,2 fucosylated H type 1 antigen present in secretors [6]. These results suggest that different P[8] strains, including the vaccine Rotarix, may differ in terms of secretor mediated susceptibility, which has also been suggested in epidemiological studies [7,8]. Moreover, a recent study suggested that the strains of the emerging lineage P[8]-4 more readily infect non-secretors compared to other P[8] strains [8], further demonstrating that there can be a difference in secretor-specificity between lineages of the same P-genotype.

To conclude, there is strong evidence that secretor status is important for susceptibility to rotavirus in a P-genotype-dependent manner, with secretors more susceptible to P[8] and P[4] infections compared to non-secretors [2]. Infection with the P[6]-genotype is strongly associated with Lewis negativity, independent of secretor status. However, variations in secretor-specificity has been observed between studies, as well as between and within lineages of P[8]-genotypes, including the Rotarix vaccine [2,7,8]. 

## 2. The Biosynthesis Pathway Determining Secretor Status

HBGAs are synthesized by stepwise addition of monosaccharides to precursors. This process is catalyzed by glycosyltransferases encoded by secretor (*FUT2*), Lewis (*FUT3*), and *ABO* genes. The FUT2 enzyme is mostly active in epithelial tissues, and HBGAs under the control of FUT2 are therefore present mainly in the mucosa of the genitourinary, respiratory, and digestive tracts, as well as free oligosaccharides in body fluids such as saliva, tears, and breastmilk. The FUT2 enzyme forms the H type 1 antigen by addition of an α1,2 linked fucose to the H type 1 precursor. The H type 1 antigen can subsequently act as a substrate for the A or B enzymes encoded by the *ABO* gene which add either an acetylgalactosamine or a galactose, respectively, to the H type 1 antigen. Secretor status is determined by whether the FUT2 enzyme is functional or not. Individuals with a non-functional FUT2 enzyme lack α1,2-linked fucose HBGAs in the epithelial mucosa and other secretions and are termed non-secretors. Secretors are individuals with a functional FUT2 enzyme that is able to express and secrete α1,2-linked fucose HBGAs, such a H-type 1 and blood group antigens. The Lewis antigens are synthesized with the FUT3 enzyme, which adds a α1,4-linked fucose residue on the H antigen precursor (non-secretor) or the H type 1 antigen (secretor), generating Lewis a or Lewis b antigens, respectively. Individuals with inactive FUT3 enzyme do not express Lewis a and b, and are termed Lewis-negative [2,9,10].

## 3. Global Distribution of Secretor Status

The distribution of HBGAs, including secretor, Lewis, and ABO pheno/genotypes, is strongly dependent on ethnicity and thus varies largely between populations and geographic locations. As secretor and other HBGAs are associated with susceptibility to many infectious diseases, this is likely a strong evolutionary driver for this observed diversity [10,11]. In European, North American, Central Asian, and several African populations, secretors constitute approximately 75–80% of the population, whereas secretor prevalence in South American populations is approximately 85–90% [12,13,14]; and notably, in Mesoamerican populations, it is as high as 90–95% [13]. In contrast, in other locally specific populations, for example, in Saudi Arabia and the Philippines, the non-secretor phenotype can constitute upwards of 50% of the population [15]. It is further important to consider that secretor status is not binary; there are several missense mutations in the *FUT2* gene that result not in inactivation of the FUT2 enzyme, but to reduced fucosyltransferase activity; and other factors, such as microbiota, can also affect glycosylation in the gut [10,16]. For example, in several East Asian populations, the non-secretor phenotype is rare. Instead, the weak-secretor genotype, which renders a low, but not absent, expression of secretor glycans, has approximately 15–20% prevalence [10,17]. 

In regard to the Lewis-negative phenotype, resulting from an inactive FUT3 enzyme, there are also significant differences between populations and geographic regions. The Lewis-negative phenotype is much more prevalent in several African countries and/or ethnicities (20–35%) when compared to European, North American, and several Asian populations (6–11%) [2,10,18].

### Relevance for Vaccine Take 

These geographical differences, in regard to HBGA expression, coupled with rotavirus infection susceptibility in a P-genotype dependent manner, can further drive global epidemiological patterns of circulating rotavirus genotypes. As secretors predominate in most populations, these populations would be more susceptible to genotypes more readily infecting secretors such as P[8] and P[4]. A striking example is the global prevalence of the P[6] genotype, which is common in Sub-Saharan Africa, and to some extent in Southeast Asia, but is almost completely absent elsewhere [2]. The proportion of this genotype associates well with the proportion of the Lewis-negative phenotype, the susceptibility marker for P[6] infection, which is vastly more common in Sub-Saharan African populations [2,19,20].

## 4. Secretor Status and Susceptibility to the Live Rotavirus Vaccines

The live oral attenuated rotavirus vaccines have successfully reduced the mortality and morbidity of rotavirus disease worldwide [21]. However, the efficacy varies considerably between regions, with low-income countries, particularly in Sub-Saharan Africa and Asia, demonstrating lower vaccine efficacy [21,22]. Several reasons have been proposed for the difference in efficacy, including early rotavirus infections, transplacental antibodies, concomitant infections, enteric dysfunction, nutritional status, and gut microbiota [22,23]. 

The two most widely used rotavirus vaccines, Rotarix and RotaTeq, are oral live attenuated vaccines containing the predominant P[8] VP4 genotype. Thus, as for natural infections, it was hypothesized that non-secretor infants would have a suboptimal immune response with Rotarix or RotaTeq due to resistance to the live virus to infect and replicate. 

Indeed, several studies have tested this hypothesis by measuring IgA seroconversion and/or vaccine strain shedding in association with secretor status and other HBGAs (Table 1). Most studies have investigated Rotarix, whereas limited studies exist for other vaccines, such as RotaTeq and the neonatal vaccine candidate RV3-BB. Most studies for Rotarix have reported a significantly lower proportion of both seroconversion and vaccine shedding in non-secretors as compared to secretors (Table 1). For RotaTeq, studies from Nicaragua reported that secretor-negative and Lewis-positive (Lewis a-phenotype) had less vaccine shedding, as well as seroconversion, but more studies from other populations are needed (Table 1). A study from New Zealand reported that the neonatal vaccine candidate RV3-BB (P[6] genotype) had a good vaccine take in infants independently of secretor and Lewis status [24]. Cumulative vaccine take was measured, defined as seroconversion and/or vaccine strain shedding after any of the three doses given, which might influence interpretation of the results (Table 1); and more studies, including measuring after one dose to limit the influence of natural early infections, are needed. Moreover, studies on secretor status and susceptibility to other globally or locally licensed vaccines and vaccine candidates, several of other P-genotypes, such as Rotavac (G9P[11]), Rotavin-M1 (G1P[8]), Lanzhou (G10P[12]) RV3-BB (G3P[6]), and Rotasiil (a combination of human G genotypes and a bovine strain of P-genotype P[5]), are lacking [25,26].

### Relevance for Protection to Rotavirus Disease

Studies thus strongly suggest that secretor status affects vaccine take to the Rotarix vaccine, with non-secretor children having less immunological response after vaccination. Important to consider is whether this also translates to vaccine failures, as children resistant to the live vaccines would also be resistant to naturally circulating rotavirus genotypes of the same secretor specificity. The degree of vaccine failures likely depends on the HBGA-phenotype proportions in the population and degree of circulation of rotavirus strains able to infect non-secretors. For example, P[6] strains have been observed to infect independently of secretors status. However, these strains predominately infect Lewis-negative children, a phenotype uncommon outside of Sub-Saharan Africa and some parts of Asia. A recent study also suggested that the emerging P[8]-4 strain more readily infects non-secretors compared to other P[8] lineages [8].

A few previous studies in highly vaccinated populations have shown that non-secretors had reduced the risk of rotavirus gastroenteritis, due to natural resistance to naturally occurring genotypes [30,33,34], thus suggesting that, even if non-secretors have lower vaccine take, their natural resistance to wild-type infections counterbalances this. 

To conclude, more studies from different geographic setups are needed to address whether, and to what extent, proportion of non-secretor children will translate to vaccine failures. Moreover, most studies have addressed the Rotarix vaccine, and studies with other orally administered live vaccines are for their efficacy in association to secretor phenotype are currently lacking. 

Important factors that possibly can determine the extent, if any, that secretor status affects rotavirus disease in vaccinated populations are likely both the proportion of non-secretors and Lewis-negative individuals, as well as the prevalence of secretor independent rotavirus genotypes such as P[6] and putatively the P[8]-4 lineage.

## 5. Secretor Status and Levels of Maternal Rotavirus Immunoglobulins

Several studies have investigated rotavirus-specific antibody titers in adults in association with secretor status [9,35,36,37,38]. All of these studies have shown that secretors have higher serum rotavirus IgG titers, as well as higher neutralization antibody titers against the predominant strains in the study population (mostly P[8]). Other studies have also observed higher salivary rotavirus IgA titers [24] in secretors, as compared to non-secretors. This higher rotavirus antibody titer in secretors likely reflects a larger number of previous infections driven by a larger greater genetic susceptibility. The available body of evidence thus clearly demonstrates an impact of secretor status on the levels of different rotavirus-specific immunoglobulins in adults. 

### Relevance for Vaccine Take

A proposed factor for the reduced rotavirus vaccine efficacy observed in some settings is through interference of maternal antibodies. The efficacy of the currently orally administered live-attenuated vaccines may be impacted both due to the placentally transferred IgG immunoglobulins and the mucosal IgA through breastmilk [39,40]. Several studies, mostly in low- and middle-income (LMIC) countries, have addressed these questions and observed a correlation between higher maternal antibody titers both in serum and breastmilk, and reduced rotavirus vaccine response [40,41,42]. In addition, an earlier study investigating rotavirus specific neutralization titers in mothers and children less than 2 years of age reported that low serotype-specific neutralizing antibody titers in the mother predispose the child for infection to the same serotype [43]. Moreover, higher levels of neutralizing antibodies in breastmilk in LMIC were shown to correlate with lower rotavirus seroconversion rates in their children undergoing vaccination [39]. 

Thus, as secretor status has been shown to influence rotavirus antibody titers, the maternal secretor status might influence the likelihood of vaccine interference through maternal antibodies. However, few studies have addressed this question directly. One such study from Bangladesh observed that breastfed children whose mothers were secretors had less IgA seroconversion rates after vaccination, compared to children whose mothers were non-secretors [44], with the infant serum rotavirus IgG titers being higher in the children of secretor mothers. More studies are thus needed to assess to which degree maternal secretor status and, thus, maternal antibody levels affect the rotavirus vaccine response in infants. An alternative to avoid suboptimal immune response due to interference by transplacental antibodies during vaccination could be using a higher vaccine dose, as reported previously [45].

## 6. Secretor Status, Breastmilk Composition, and Infant Microbiota

The gut microbiota have been indicated to be important for susceptibility to enteric viruses in several studies through several mechanisms, such as enhancing viral attachment, promoting immune evasion, and affecting epithelial glycosylation [16,36,46]. Microbiota depletion has further been observed to reduce rotavirus infection in a mice model [47]. Blood-group antigen expression has been associated with differences in gut microbiota composition of adults [48,49,50]. Studies on secretor status and microbiota composition, however, demonstrate conflicting results. Although some studies have found secretor status to influence microbiota composition, often leading to a difference in diversity of bacterial population [16,36,51,52], other studies have found no or weak associations [53,54,55,56].

More relevant for vaccine take, however, is the infant microbiota. The intestinal microbiota of infants is very different from adults and shows important individual variability [57]. The mother is an important external factor for the development of the infant microbiota, due to contacts during birth, nursing, and early feeding [57]. While factors such as skin contact are important, a large influence on infant microbiota is breastmilk [58]. In addition to other nutrients, breastmilk contains a mixture of complex human milk oligosaccharides (HMOs), which can function as a growth substrate [58]. In addition to this prebiotic nature of HMOs, breastmilk has probiotic properties that help to shape the infant gut microbiota. During the first months of life, when breastmilk is typically the sole source of nutrition, the gut bacteria present vary significantly depending on the mother [58].

The composition of breastmilk HMO is significantly influenced by secretor status [59,60], where HMOs of secretor positive mothers contain α1–2 fucosylated HMOs—predominantly 2′fucosyllactose (2′FL), and, to a lesser extent, lactodifucotetraose, lacto-N-difucohexaose I, and lacto-*N*-fucopentaose I. Non-secretor mothers lack or have only traces of these in breastmilk, as well as a lower total amount of HMOs [61,62]. These differences in HMO composition influence the infant’s microbiota [63]. As an example, several studies have shown that positive secretor status in the mother is associated with a higher abundance of bifidobacteria in infant microbiota [63,64]. These associations are, however, not universal, and a negative correlation between 2′FL in breastmilk and relative abundance of bifidobacteria in infant feces was observed in another study [65]. Besides maternal secretor status, environment and geographic location could also affect varied HMO compositions between women [66]. This suggests that the effect of fucosylated HMOs on infant microbiota is different between groups, depending on other factors, such as environment and geographic location [61,66].

### 6.1. Relevance for Vaccine Take

#### 6.1.1. Microbiota

The interactions between virus, host, glycans, and microbiota are complex [16]. Secretor status of the mother is important for determining infant microbiota composition at time of vaccination. While studies have indicated a link between microbiota and vaccine take for other oral vaccines [67], and probiotic supplementation has been associated with a modest increase of Rotarix immunogenicity in a randomized control trial [68], a clear link between microbiota and rotavirus vaccine response remains elusive. Studies from Ghana and Pakistan with the Rotarix vaccine observed a difference in microbiota composition between responders and non-responders [69,70]. However, studies from Nicaragua with RotaTeq vaccine and from India with the Rotarix vaccine did not observe any association [71,72]. Moreover, as secretor status influences both microbiota composition and factors that can be important for rotavirus vaccine take (Figure 1), it can be difficult to rule out confounding factors and ascertain the specific role of microbiota in these studies.

#### 6.1.2. Non-Antibody Components of Breastmilk

Apart from rotavirus IgA titers in breastmilk, non-antibody components of breastmilk may also directly affect vaccine take in a secretor-dependent manner, both by facilitating or restricting vaccine strain infection, depending on type of rotavirus vaccine or administration schedule [44,73,74]. Fucosylated glycans in milk of secretor mothers have been reported to provide protection to diarrhea in breastfed infants [75]. It has been further observed that both the breastmilk profile and the inhibitory effect on rotavirus vaccines due to non-antibody components in the breastmilk vary among developed and developing countries [76,77]. Dose-dependent increase in infectivity of a neonatal RV vaccine candidate (I321, G10P[11]) with specific HMOs [73] leads to speculation that this might be also true for the other licensed G9P[11] vaccine (Rotavac) or for the RV3-BB vaccine, which has also shown a good vaccine take when given to children at day 0 or birth. 

Moreover, fucosylated HMOs in breastmilk of secretor mothers have been shown to inhibit the in vitro infectivity of P[8] and P[4] rotaviruses, the most prevalent global rotavirus strains [74]. This is suggestive that similar processes may be occurring in vivo after oral vaccination with Rotarix. These findings may be attributable to fucosylated glycans acting as decoy receptors for vaccine [44], restricting vaccine take (Figure 1). 

Breastfed children in Bangladesh whose mothers were secretors had less IgA seroconversion rates after Rotarix vaccination, compared to children whose mothers were non-secretors, controlling for infant’s secretor status [44]. The breastmilk further contained similar levels of rotavirus IgA, thus suggesting a secretor-dependent non-antibody effect of the breastmilk on rotavirus vaccine take. However, several studies involving restriction of breastfeeding at time of vaccination have not observed enhancement of the IgA immune response to oral rotavirus vaccines in children [40,48,78].

#### 6.1.3. Conclusions

To conclude, breastmilk composition is affected by secretor status of the mother, leading to higher presence of fucosylated glycans, which in turn affect microbiota composition in the infant gut (Figure 1). Although both microbiota composition and the presence or absence of fucosylated glycans in breastmilk have been postulated to affect the rotavirus vaccine take, more studies are needed to clearly ascertain a link to vaccine take of different rotavirus vaccines.

## 7. Concluding Remarks and Perspectives

The available literature shows that secretor status affects several factors that might be important for an optimal rotavirus vaccine take in infants. Firstly, secretor status of the infant is associated with innate susceptibility to infection of some of the live rotavirus vaccines, important for subsequent immune response (Figure 1). Most studies have been performed in regard to Rotarix, and less is known of the effect on secretor status and susceptibility to other live oral rotavirus vaccines; however, there are indications that secretor status does not affect the vaccine take of the neonatal RV3-BB vaccine of P[6] genotype. Secretor status also affects the rotavirus antibody specific titers in adults which can restrict vaccine take in the infant at time of vaccination through interference of maternal antibodies. Secretor status further influences maternal breastmilk composition, which in turn can affect gut microbiota composition in the infant. However, clear evidence between infant microbiota composition and rotavirus vaccine take is lacking and warrants more studies. The fucosylated non-antibody components in breastmilk of secretors may also have a stronger antiviral effect on common rotavirus strains, including the predominant vaccines, which could thus also restrict vaccine take (Figure 1). However, studies on restriction of breastfeeding at time of vaccination have not yielded any clear differences in terms of rotavirus vaccine response.

Moreover, several of the recently available oral rotavirus vaccines have been limited or not investigated in regard to the association between secretor status and vaccine take. As these vaccines can consist of other P-genotypes, and/or are neonatally administrated, the impact of secretor status on vaccine take will likely differ.

The interaction between secretor status in both mother and infant in association with these and other factors is complex and warrants further investigation. Importantly, there is a paucity in studies accounting for the secretor status of both the infant and the mother, considering all factors that are affected by secretor status. Moreover, several of these factors, while restricting rotavirus vaccine take, will also provide protection of the infant to naturally circulating rotavirus and thus symptomatic disease. Non-secretor infants can be protected against natural circulating strains, and higher levels of transplacental antibodies and anti-rotavirus effect of breastmilk in secretor–secretor mothers will also provide protection of the infant to rotavirus infection.

Thus, it is of importance to assess not only the secretor-status-mediated effects on vaccine response, but also on protection towards symptomatic rotavirus disease in longitudinal studies in vaccinated populations.

## Figures and Tables

**Figure 1 viruses-13-01144-f001:**
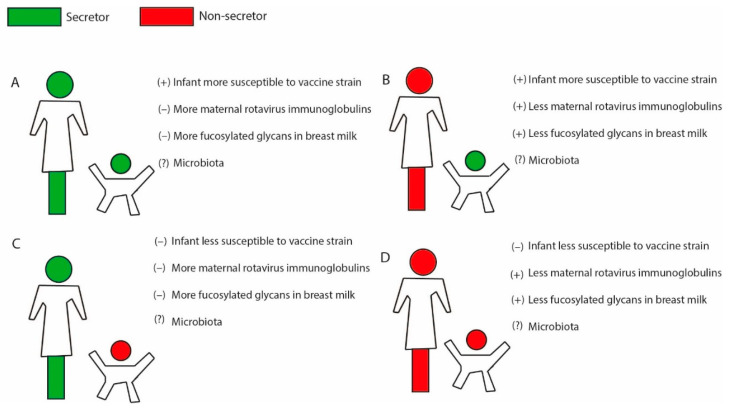
A simplistic overview on how secretor status in both mother and infant affects several factors that, in turn, may restrict rotavirus vaccine take. (**A**) Both mother and infant are secretors. In such a scenario, the positive secretor status of the child favors susceptibility to live oral rotavirus vaccines with a P[8] component such as Rotarix and RotaTeq, while higher levels of rotavirus-specific maternal antibodies (IgG, IgA) and anti-rotavirus effect by IgA and/or fucosylated glycans in breastmilk might limit vaccine take. (**B**) Mother is non-secretor, and infant is a secretor. Here, the secretor-positive infant will be more susceptible to live oral rotavirus vaccines (Rotarix and RotaTeq), while lower rotavirus-specific maternal antibodies and lower of levels of IgA and/or fucosylated glycans in breastmilk may also increase vaccine take. (**C**) Mother is a secretor and infant is a non-secretor. The infant will be less susceptible to live oral rotavirus vaccines due to negative secretor status. Higher levels of maternal rotavirus antibodies and anti-rotavirus effect of breastmilk might further restrict vaccine take. (**D**) Both mother and infant are non-secretors. Infant will be less susceptible to live oral rotavirus vaccine, while lower levels of maternal rotavirus antibodies and lower anti-rotavirus effect of breastmilk might benefit vaccine take. The different scenarios will also influence infant microbiota composition, but specific effects on rotavirus vaccine take are unclear. Following this model, it is proposed that scenario B will be most beneficial in terms of vaccine take, while scenario C will be the least beneficial. It is important to note that the same factors restricting vaccine take also provide protection of the infant to natural rotavirus infections. Note: (+) and (–) indicate higher and lower vaccine take, respectively, while (?) represents unknown effect.

**Table 1 viruses-13-01144-t001:** Summary of different studies investigating the association between rotavirus vaccine take and secretor and Lewis status. Updated and modified from Reference [2].

Vaccine	Country	Secretor Status	Lewis Phenotype	Measurement	Reference
				Seroconversion	
Rotarix	Nicaragua	Non-secretors less seroconversion	Lewis A no seroconversion	After 1 dose	[27]
	Pakistan	Non-secretors less seroconversion	No association	After 3 doses	[28]
	Ghana	Non-secretors less seroconversion	No association	After 2–3 doses	[29]
	Malawi	Non-secretors less seroconversion	No association	After 2 doses	[30]
RotaTeq	Nicaragua	No association	Lewis A no seroconversion	After 1 dose	[27]
RV3-BB	New Zealand	No association	No association	Cumulative	[24]
				Vaccine shedding	
Rotarix	Nicaragua	Non-secretors no shedding	No shedding in Lewis A	After 1 dose	[31]
	Malawi	Non-secretors less shedding	No association	After 1 dose	[30]
	Malawi	No association	No association	After 2 doses	[30]
	South Africa	Non-secretors less shedding	Lower shedding in Lewis A	After 1 dose	[32]
	Brazil	Non-secretors less shedding	Lower shedding in Lewis A	After 1 dose	[7]
RotaTeq	Nicaragua	No association	No shedding in Lewis A	After 1 dose	[31]
RV3-BB	New Zealand	No association	No association	Cumulative	[24]

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
