# Peer review of "Effect of Infant and Maternal Secretor Status on Rotavirus Vaccine Take—An Overview"

_viruses, 2021, doi:10.3390/v13061144_

Round 1

Reviewer 1 Report

The review is well written in a clear, informative, and objective manner. The manuscript reviews available data on the already known and the still unknown factors possibly involved in virus susceptibility and vaccine take interference which may influence on degree of vaccine efficacy.

My only suggestion is that the authors include in the review a paragraph on the influence of enzymes, such as trypsin-like enzymes, and possibly other factors, such as ions and soluble factors in the lumen millie that may also play a role in susceptibility/resistance to rotavirus infection and possibly on vaccine take.

Author Response

We thank the reviewer for the comments and suggestions.

We agree with the reviewer that also enzymes as well as other host factors not currently mentioned in the review could affect rotavirus susceptibility and thus also potentially vaccine take.

However, we were unable to find any information linking these other potential host factors, such as trypsin-like enzymes, to secretor status in humans. As the effects of secretor status was the focus of this review, we decided only to include factors to which, to our knowledge, an effect/association to secretor status had been described. 

Reviewer 2 Report

This is a very well-written and comprehensive review of the effects of infant and maternal secretor status on rotavirus infection and responses to oral rotavirus vaccines. While there have been previous reviews on the topic of HBGAs and their impact on rotavirus diarrhoea and vaccine take, this review provides additional information on the role of maternal secretor status in susceptibility to rotavirus disease and vaccine take in their infants. The inclusion of a discussion on the microbiome in relation to this is also important. I do not have any major comments. There are a few minor editorial comments which can be addressed during the journal editing phase rather than raising these as specific comments to the authors.

  • “known and unresolved effects” – consider changing to “known and possible” and just saying effects.
  • Line 23 – “cause” rather than agent.
  • Line 25 – add five years
  • Correction of minor grammatical errors e.g. line 233 – “Non-secretor mothers lack”; Line 335 – “Non-secretor infants can be protected against…”

Author Response

We thank the reviewer for the comments and suggestions.

All the suggestions from the revewier have been adressed and marked with track changes in the manuscript.

Reviewer 3 Report

This paper entitled “Effect of infant and maternal secretor status on rotavirus vaccine take – an overview” is a review of an interesting and controversial topic, the influence of the secretor status of both mother and infant on the protective effect of rotavirus vaccines on the latter.

The authors have exhaustively revised the literature and 78 references have been quoted. The review is rather complete, as not only the secretor status, but also other relevant and linked aspects like the mother breast milk composition, the infant gut microbiota, antibody and several non-antibody components of breast milk that were investigated in different studies are analysed.

Unfortunately, most reviewed studies on rotavirus vaccine take and infant and mother secretor status refer to Rotarix vaccine, a few to RotaTeq and RV3-BB and none to other licensed vaccines like Rotasiil, Rotavac, Rotavin-M1 or Lanzhou. This is a limitation of the present review, which is not attributable to the review itself, but to the lack of studies with these latest vaccines.

The paper is well written, the English language is fine and its reading is pleasant. 

Minor comments:

The following typos should be corrected:

Line 148:  invesitigating  > investigating

Line 171: …,in any,  >  …,if any,

Line 237: bifidobacterial  >  bifidobacteria

Author Response

We thank the reviewer for the comments and suggestions.

All the suggestions by the reviewer have been adressed and marked with track changes in the revised manuscript.